# Molecular Drivers of Multiple and Elevated Resistance to Insecticides in a Population of the Malaria Vector *Anopheles gambiae* in Agriculture Hotspot of West Cameroon

**DOI:** 10.3390/genes13071206

**Published:** 2022-07-06

**Authors:** Arnaud Tepa, Jonas A. Kengne-Ouafo, Valdi S. Djova, Magellan Tchouakui, Leon M. J. Mugenzi, Rousseau Djouaka, Constant A. Pieme, Charles S. Wondji

**Affiliations:** 1Medical Entomology Department, Centre for Research in Infectious Diseases (CRID), Yaoundé P.O. Box 13591, Cameroon; jonas.kengne@crid-cam.net (J.A.K.-O.); valdi.djova@crid-cam.net (V.S.D.); magellan.tchouakui@crid-cam.net (M.T.); leon.mugenzi@crid-cam.net (L.M.J.M.); 2Department of Biochemistry, Faculty of Medicine and Biomedical Sciences, University of Yaounde 1, Yaoundé P.O. Box 1364, Cameroon; apieme@yahoo.fr; 3Department of Biochemistry, University of Bamenda, Bambili P.O. Box 39, Cameroon; 4International Institute of Tropical Agriculture, Cotonou P.O. Box 0932, Benin; r.djouaka@cgiar.org; 5International Institute of Tropical Agriculture, Yaoundé P.O. Box 2008, Cameroon; 6Vector Biology Department, Liverpool School of Tropical Medicine, Pembroke Place, Liverpool L3 5QA, UK

**Keywords:** *Anopheles gambiae*, Cameroon, malaria, pyrethroids, resistance escalation, cytochrome P450s

## Abstract

(1) Background: Malaria remains a global public health problem. Unfortunately, the resistance of malaria vectors to commonly used insecticides threatens disease control and elimination efforts. Field mosquitoes have been shown to survive upon exposure to high insecticide concentrations. The molecular mechanisms driving this pronounced resistance remain poorly understood. Here, we elucidated the pattern of resistance escalation in the main malaria vector *Anopheles gambiae* in a pesticide-driven agricultural hotspot in Cameroon and its impact on vector control tools; (2) Methods: Larval stages and indoor blood-fed female mosquitoes (F_0_) were collected in Mangoum in May and November and forced to lay eggs; the emerged mosquitoes were used for WHO tube, synergist and cone tests. Molecular identification was performed using SINE PCR, whereas TaqMan-based PCR was used for genotyping of L1014F/S and N1575Y *kdr* and the G119S-*ACE1* resistance markers. The transcription profile of candidate resistance genes was performed using qRT-PCR methods. Characterization of the breeding water and soil from Mangoum was achieved using the HPLC technique; (3) Results: *An. gambiae* s.s. was the only species in Mangoum with 4.10% infection with *Plasmodium.* These mosquitoes were resistant to all the four classes of insecticides with mortality rates <7% for pyrethroids and DDT and <54% for carbamates and organophophates. This population also exhibited high resistance intensity to pyrethroids (permethrin, alpha-cypermethrin and deltamethrin) after exposure to 5× and 10× discriminating doses. Synergist assays with PBO revealed only a partial recovery of susceptibility to permethrin, alpha-cypermethrin and deltamethrin. Only PBO-based nets (Olyset plus and permaNet 3.0) and Royal Guard showed an optimal efficacy. A high amount of alpha-cypermethrin was detected in breeding sites (5.16-fold LOD) suggesting ongoing selection from agricultural pesticides. The 1014F-kdr allele was fixed (100%) whereas the 1575Y-kdr (37.5%) and the 119S Ace-1^R^ (51.1%) were moderately present. Elevated expression of P450s, respectively, in permethrin and deltamethrin resistant mosquitoes *[CYP6M2 (10 and 34-fold)*, *CYP6Z1(17 and 29-fold)*, *CYP6Z2 (13 and 65-fold)*, *CYP9K1 (13 and 87-fold)]* supports their role in the observed resistance besides other mechanisms including chemosensory genes as SAP1 (28 and 13-fold), SAP2 (5 and 5-fold), SAP3 (24 and 8-fold) and cuticular genes as *CYP4G16* (*6 and 8-fold*) and *CYP4G17* (*5 and 27-fold*). However, these candidate genes were not associated with resistance escalation as the expression levels did not differ significantly between 1×, 5× and 10× surviving mosquitoes; (4) Conclusions: Intensive and multiple resistance is being selected in malaria vectors from a pesticide-based agricultural hotspot of Cameroon leading to loss in the efficacy of pyrethroid-only nets. Further studies are needed to decipher the molecular basis underlying such resistance escalation to better assess its impact on control interventions.

## 1. Introduction

Malaria remains a global public health problem with about 228 million cases worldwide and 213 million cases (93%) recorded in Africa [1]. Scale-up of insecticide-based interventions using long-lasting insecticide nets (LLINs) and indoor residual spraying (IRS) in Africa have contributed significantly to the reduction in malaria cases in the recent years [2]. Pyrethroids are the main insecticide class approved for LLINs impregnation, as well as the most common insecticide class used in IRS [1]. Unfortunately, the high selective pressure caused by the intensive use of pesticides threatens the effectiveness of malaria control and elimination efforts.

Between 2013 and 2014, insecticide resistance assessment followed the WHO protocol published in 2013 which recommended exposing insects to predefined discriminating doses and reporting mortality thereafter [3]. However, it soon became apparent that this methodology was limited in really alerting to the impact of observed resistance on malaria control tools. In Burkina Faso, the comparative analysis of resistance data based either on discriminant doses or on the assessment of resistance intensity showed that the latter provides a better overview of the impact of resistance on malaria control tools [4,5]. Up to a 10-fold increase in the level of resistance to pyrethroids was noted in *An. gambiae* collected in Vallée du Nkou in Southwest Burkina Faso over a short period of one year [5]. This resistance escalation could have a drastic impact on the development of multiple resistance to other insecticide classes and negatively impact the success of current and future insecticide-based interventions. Between 2009 and 2014, there was an increase in resistance to pyrethroids and carbamates in *Anopheles funestus* from Malawi, which resulted, in 2014, in the incidence of multiple resistance including Organochlorines (DDT and dieldrin) [6]. This observation suggests that the rise in resistance to an insecticide because of selective pressure on insect populations could be at the origin of the emanation of new mosquito populations capable of resisting several other classes of insecticides due to potential cross-resistance mechanisms between these insecticide classes. In the same line, there is evidence that the use of organic fertilizers in areas of intense agricultural activities, use of pesticides in agricultural areas in addition to the selective pressure exerted by vector control interventions, increase resistance to insecticides and generate multiple resistance in adult mosquitoes. In an experimental study, it was shown that adults emerging from *Anopheles arabiensis* larvae exposed to organic fertilizers had a high level of resistance to deltamethrin and permethrin, while those emerging from unexposed larvae had a low and moderate level of resistance to deltamethrin and permethrin respectively [7]. Other field studies have also shown that agricultural areas present a high risk of insecticide resistance, such as Tambacounda in Senegal [8], Dabou and Tiassalé in Côte d’Ivoire [9] and Nkolandom in Cameroon [10]. This phenomenon of multiple and elevated resistance is all the more alarming as the last WHO report on malaria noted that several countries in Asia and Africa are already experiencing multiple resistance to all four classes of commonly used insecticides [11]. Early identification of resistance escalation in the field would be an alternative to limit the effect of the emergence of these super-resistant mosquitoes (mosquitoes surviving higher doses of insecticides) on malaria control tools.

Though the mechanism of resistance escalation is not yet well-understood, some studies have shown the involvement of some mosquito’s genes in the process in many countries. In Burkina Faso, several detoxification genes in *An. gambiae*, including *CYP4G16*, *CYP9J5*, *CYP9M1*, *COEAE3G* and *GSTE5*, have been identified as potential drivers of escalation. Other cuticular genes and enzymes, such as *CPR 73*, *CPAPA3-A1a*, *CPAPA3-A1b*, the chymotrypsin-1, aquaporin and ATP synthase, were also implicated [12]. In Malawi, *CYP6M7* was also associated with resistance escalation in *An. funestus* [6]. Although *CYP9K1*, *CYP6P9a* and *CYP6P9b* have been associated with deltamethrin resistance in *An. funestus* from Uganda, no evidence for their role in resistance escalation in this locality has been found [13]. Recently, the potential involvement of three candidate resistance genes (*CYP6M2*, *CYP6P3* and *GSTD3*) on resistance escalation in *An. gambiae* in Ghana was investigated. No significant differences in the expression of these genes were observed between susceptible and resistant or super-resistant mosquitoes [14]. Much remains to be done to elucidate the molecular basis of resistance escalation in the two major malaria vectors. The present study, therefore, aims at assessing resistance intensity in an agriculture hotspot in Western Cameroon and the contribution of known resistance genes in the development of such a phenomenon. Knowledge generated here could help better inform control programs and farmers or better manage new and future insecticides between both sectors.

## 2. Materials and Methods

### 2.1. Study Site and Samples Collection

Indoor blood-fed *An. gambiae* s.l. mosquitoes were collected in Mangoum (5°29′09.2″ N 10°35′20.8″ E) situated in the western region of Cameroon at 1054 m altitude from the sea level. Mangoum is characterized by extensive manual and mechanized agricultural settings producing spices, vegetables and cereals (Appendix A). There are four seasons in this area: two rainy seasons (from March to June and from September to November) and two dry seasons (from December to February and from July to August).

Adult and larval collections were conducted in November 2020 and May 2021 targeting the beginning of the dry season and the rainy season, respectively. Adult female blood-fed mosquitoes were collected indoors (on the walls and the roof of different houses across the village between 6:00 AM and 10:00 AM) using electric aspirators (Prokopack Aspirator, Model 1419, John W. Hock Company, Gainesville, FL, USA). Mosquitoes were transported to the insectary of the Centre for Research in Infectious Diseases (CRID) in Yaoundé where they were morphologically identified and sorted by species according to the morphological identification keys of Gillies and De Meillon [15] and Gillies and Coetzee [16]. *An. gambiae* mosquito larvae were collected from ponds randomly selected from Mangoum. A 35 mL dipper was used to collect larvae from the water. The larvae were collected in basins and then transferred with a small pipette into bottles for easy transport to the laboratory. Mosquitoes breeding water and soil were sampled in November from the farms in IRAD and Djincha, two locations in Mangoum.

### 2.2. Mosquito Rearing and Molecular Identification

Eggs were obtained from collected blood-fed adult F_0_ using the tube forced oviposition method as developed by Morgan et al. [17]. After egg hatching, F_1_ larvae were fed with Tetramin™ (Tetra GmbH, Herrenteich 78, Melle, Germany) baby fish food. Both F_0_ (from the field) and F1 (from forced oviposition) larvae were allowed to develop into pupae then adult mosquitoes. Upon emergence, the F_0_ and F_1_ adult mosquitoes were kept in different cages for subsequent experiments. Female mosquitoes (randomly selected) that successfully laid eggs were dissected, and the heads/thoraces separated from the abdomens. Genomic DNA was extracted from heads/thoraces of field-collected adult mosquitoes (F_0_) using the Livak protocol [18]. Identification of species within *An. gambiae* s.l. was performed using the short interspersed elements (SINE) PCR protocol [19].

### 2.3. Determination of Pesticide Residues in Mosquito Breeding Water and Soil Samples

Based on the most common agrochemicals used by the farmers, two compounds were screened: deltamethrin and alpha-cypermethrin (type II pyrethroid insecticides). Two replicates of 5 g of soil samples and three replicates of 50–500 mL of breeding-site water samples were collected following the method previously described by Djouaka et al. (2018) [20]. Solid phase extraction (SPE) was performed as described previously by Guan and Chai (2011) [21], using Supelco Visiprep™ (Merck, Germany) dispositive for separation, purification and pre-concentration of insecticide residues before the HPLC quantification. Briefly, the SPE solid sorbent (column) was activated by methanol followed by addition of 30 mL of sample. The next step consisted of eluting the insecticide residues using acetonitrile solvent for HPLC gradient grade (≥99.9% purity, CHROMASOLV™ of Honeywell brands Riedel-de Haën™ from Germany), then the pre-concentration of insecticide residues was achieved by evaporation of acetonitrile until a final volume of 1 mL that was used for HPLC detection and quantification of insecticide residues in the sample. The samples were quantified alongside two standard solutions of deltamethrin (purity ≥ 98.0% from Sigma-Aldrich, Merck, UK) and alpha-cypermethrin (purity ≥ 98.0% from Sigma-Aldrich, Merck, UK). Before injection, a two-fold dilution of the standards was performed (with concentrations ranging from 0.625 μg/mL to 40 μg/mL). The injections of 20 μL of each standard concentration allowed the generation of calibration curve from which the linear regression equation was obtained as follows: y = ax + b (y represents the area of the peak; x, the concentration (µg/mL); where “a” and “b” are constants. The detection limit of the method was determined using the ICH guideline for the detection limit parameters of the analytical method validation as described previously by Shrivastava and Gupta, (2011) [22]. The limit of detection (LOD) and limit of quantification (LOQ) can be expressed as LOD = 3σ/a and LOQ = 10σ/a, where σ is the standard deviation of the response and a is the slope of the calibration curve. HPLC analysis was performed with a reverse phase HPLC machine Agilent technology 1260 infinity (Agilent Technologies Deutschland GmbH & Co. KG, Waldbronn, Germany). The HPLC column used was C18, 5 µm 120 Â, 4.6 × 250 mm (Thermo Fisher scientific, Waltham, MA, USA).

### 2.4. Estimation of Sporozoite Rate

The presence of the salivary sporozoites was investigated using a TaqMan genotyping protocol, established by Bass et al. [23]. Primers described by Bass were used, together with two probes labeled with fluorophores, FAM to detect *Plasmodium falciparum*, and HEX to detect the combination of *Plasmodium ovale*, *Plasmodium vivax* and *Plasmodium malariae* (Appendix A). Positive controls (known FAM+ and OVM+) were used, in addition to a negative control, in which 1 µL of ddH2O was added. To validate the findings of the TaqMan assay, a nested PCR of Snounou et al. [24] was carried out, using all the samples that were tested positive with TaqMan. The sporozoite rate was calculated as the percentage of positive female mosquitoes, relative to the total number of the female mosquitoes examined.

### 2.5. Susceptibility Tests and Resistance Intensity 

F_0_ and F_1_ females *An. gambiae* s.s were used for insecticide resistance monitoring. Susceptibility tests were carried out using WHO protocol for adults [25]. The four major public health insecticide classes were tested including (i) the type I pyrethroid: permethrin (0.75%); (ii) the type II pyrethroid: alpha-cypermethrin (0.05%) and deltamethrin (0.05%); (iii) the organochlorine: DDT (4%); (iv) the carbamate: bendiocarb (0.1%); and (v) the organophosphate: malathion (5%). All insecticide-impregnated papers were sourced from the WHO/Vector Control Research Unit (VCRU) of the University of Sains Malaysia (Penang, Malaysia). Four replicates of 20–25 female mosquitoes (3–4 days old) per tube were used for each insecticide. To establish the strength of pyrethroid resistance, additional bioassays were performed, with permethrin (5×: 3.75% and 10×: 7.5%), alpha-cypermethrin (5×: 0.25% and 10×: 0.5%), and deltamethrin (5×: 0.25% and 10×: 0.5%). The mosquitoes were transferred to tubes with insecticide-impregnated papers and exposed for 1 h. The number of mosquitoes knocked down by the insecticide was recorded after 1 h of exposure. Next, mosquitoes were fed with a 10% sugar solution, and the number of dead mosquitoes was recorded 24 h post-exposure. The Abbott’s formula was used to correct for control mortality when it was between 5 and 20%. Tests with untreated papers were systematically run as controls. The mosquitoes were deemed susceptible to an insecticide when mortality was >98%, suspected to be likely resistant when mortality was between 90–98% and resistant when mortality was <90% [25]. Figures were prepared using R studio 4.0 software. All mosquitoes alive and dead 24 h after the end of the bioassay were preserved in RNAlater (Qiagen) and silicagel, respectively, for molecular analyses.

### 2.6. Synergist Test

To investigate the potential role of cytochrome P450 monooxygenases in pyrethroid resistance, a synergist bioassay was carried out using 4% Piperonyl Butoxide (PBO) [an inhibitor of CYP450s [26]] against 1× permethrin (0.75%), 1× alpha-cypermethrin (0.05%) and 1× deltamethrin (0.05%). The insecticides and PBO were sourced from the WHO/Vector Control Research Unit (VCRU) of the University of Sains Malaysia (Penang, Malaysia). Four replicates of 20–25 F females (3–4 days old) were pre-exposed to PBO for 1 h, and then transferred to tubes containing permethrin, alpha-cypermethrin, and deltamethrin, respectively [25]. Mosquitoes were treated as in the WHO bioassays described above, and mortalities scored after 24 h. Two replicates of 25 females each were exposed to PBO only, as a control.

### 2.7. Determination of LLINs Efficacy with Cone Test

To investigate the efficacy of commonly distributed long-lasting insecticidal nets (LLINs), a cone test was conducted following the WHOPES protocol [27,28], using 3–4 day old female mosquitoes. Five replicates of 9–12 mosquitoes were placed in a plastic cone attached to an insecticide-containing bed net and tested. The LLINs include: the Olyset^®^ Net (Sumitomo Chemical, Tokyo, Japan; containing 2% permethrin), Olyset^®^ Plus (Sumitomo Chemical, Tokyo, Japan; containing 2% permethrin combined with 1% of the synergist, piperonyl butoxide, PBO), PermaNet^®^ 2.0 (Vestergaard Frandsen, Lausanne, Switzerland; containing 1.4–1.8 g/kg ± 25% deltamethrin), PermaNet^®^ 3.0 side (Vestergaard Frandsen, Lausanne, Switzerland; containing 4.0 g/kg ± 25% deltamethrin), PermaNet^®^ 3.0 roof (Vestergaard Frandsen, Lausanne, Switzerland; containing 4.0 g/kg ± 25% deltamethrin, combined with 25 g/kg ± 25% of PBO), Duranet^®^ (Shobikaa Impex Private Limited, Tamil Nadu, India; containing 261 mg/m^2^ alpha-cypermethrin), Interceptor^®^ (BASF Corporation, Ludwigshaven, Germany; containing 200 mg/m^2^ ± 25% alpha-cypermethrin), Interceptor G2^®^ (BASF Corporation, Ludwigshaven, Germany; containing 100 mg/m^2^ ± 25% alpha-cypermethin and 200 mg/m^2^ ± 25% chlorfenapyr) and Royal Guard^®^ (IVCC, Liverpool, England; containing 209 mg/m^2^ alpha-cypermethrin and 225 mg/m^2^ pyriproxyfen). For each standard, PBO and novel nets, five different pieces cut from an LLIN brand were used for the five technical replicates. Mosquitoes were exposed for 3 min and immediately transferred to paper cups. They were supplied with 10% sucrose and mortalities were recorded after 24 h and specially for Interceptor G2 at 48 h and 72 h since this net contains a slow-acting ingredient (chlorfenapyr). For the control, five replicates of ten mosquitoes were exposed to an untreated net.

### 2.8. Genotyping of the Target-Site Resistant Markers in An. gambiae s.s. from Mangoum

The *L1014F-kdr*, *L1014S-kdr*, and *N1575Y* mutations responsible for pyrethroid resistance, and *G119S-ACE 1* associated with carbamate and organophosphate resistance in *An. gambiae* s.s. were genotyped in Mangoum F_0_ mosquitoes using Taqman assay (Appendix A) [23]. Considering the sample size available in alive and dead mosquitoes after 1 h of exposure to permethrin and deltamethrin (1×, 5×and 10×), we also assessed the ability of mosquitoes with the 1575Y-kdr mutant allele to survive to insecticides. The reaction mixture of 10 μL final volume containing 1×Sensimix (Bioline, London, UK), 80×primer/probe mix, and 1 μL template DNA was used for this assay. The probes were labeled with two distinct fluorophores: FAM to detect the resistant allele and HEX to detect the susceptible allele. The assay was performed on an Agilent MX3005 real-time PCR machine (Agilent Technologies Germany GmbH & Co.KG, Waldbronn, Germany) with cycling conditions of 95 °C for 10 min, followed by 40 cycles at 95 °C for 15 s and 60 °C for 1 min as previously described [23]. Association between genotypes and resistant phenotype was assessed by calculating the odds ratio of alive and dead individuals between the homozygous resistant, heterozygote and homozygous susceptible individuals. Statistical significance was computed based on the Fisher’s exact probability test.

### 2.9. Polymorphism Analysis of the Voltage-Gated Sodium Channel Gene in An. gambiae from Mangoum

The genetic diversity of the voltage-gated sodium channel (VGSC) gene was investigated for *An. gambiae s.s* from Mangoum. Exon 20 of the VGSC gene (including the 1014 codon associated with kdr) was amplified in 15 field-collected females *An. gambiae* s.s. females, purified and sequenced as previously described [29,30]. Sequences were aligned using ClustalW under BioEdit v.7.2.5 [31], whereas haplotype reconstruction and polymorphism analysis were performed using DnaSPv6 [32]. Mangoum haplotypes were compared to the 5 kdr haplotypes previously detected across Africa as containing either the L1014, 1014S or the 1014F mutations [29,33,34,35].

### 2.10. Transcription Profile of Metabolic Resistance Genes in An. gambiae s.s.

The transcription patterns of 13 candidate genes previously shown to be associated with pyrethroid resistance in *An. gambiae s.s.* (*CYP4G16*, *CYP4G17*, *CYP6M2*, *CYP6Z1*, *CYP6Z2*, *CYP9K1*, *GSTe2*, *SAP1*, *SAP2*, *SAP3*, *CYP6P1*, *CYP6P3 and CYP6P4*) were assessed by a quantitative reverse transcription PCR (qRT-PCR) in permethrin 1× alive, 5× alive, 10× alive, deltamethrin 1× alive, 5× alive, 10× alive and unexposed mosquitoes relatively to the susceptible strain KISUMU and using two housekeeping genes: Elongation factor (AGAP000883) and Ribosomal Protein S7 (AGAP010592) (Appendix A). Total RNA was extracted from 3 batches of 10 mosquitoes each and similarly from the susceptible laboratory strain KISUMU, using the Arcturus PicoPure RNA isolation kit (Life Technologies, Carlsbad, CA, USA), according to the manufacturer’s instructions. cDNA (complementary Deoxyribonucleic acid) was synthesized from the purified RNA by Reverse transcriptase-PCR using the SuperScript III (Invitrogen, Waltham, MA, USA) and the oligo-dT20 and RNAse H (New England Biolabs, Ipswich, MA, USA) kit in a total reaction volume of 20 μL. Amplification was performed in Agilent Mx3005 qRT-PCR thermocycler (Santa Clara, CA, USA) with the following conditions: 95 °C for 3 min followed by 40 cycles of 10 s at 95 °C and 10 s at 60 °C; 1 min at 95 °C; 30 s at 55 °C and 30 s at 95 °C. Samples were amplified in at least two technical replicates, using three biological replicates for gene expression analysis for each population. MxPRO v. 4.10 software (Santa Clara, CA, USA) was used for the calculation of Ct values for each reaction. Standard curves of assessed genes were established. As the efficiency was different from 100%, Ct value was adjusted according to efficiency as previously performed [36]. The relative expression was calculated individually according to the 2−DDCT method [37] and plotted with R studio software. Genes were considered as relatively overexpressed with respect to Kisumu when their relative fold changes (FCs) were more than 2-fold change. Dunnett’s test was also used for comparing several treatments (Exposed) with a control (Unexposed). Fold changes were compared between the exposed groups using Kruskal-wallis test.

## 3. Results

### 3.1. Species Composition and Plasmodium Infection Rate

A total of 3289 specimens were collected indoor using electric aspirators, 668 in November 2020 and 2621 in May 2021, respectively.

DNA was extracted from 195 (94 from November and 101 from May) F_0_ blood fed mosquitoes from Mangoum. Molecular identification of 94 F_0_
*An. gambiae s.l* mosquitoes from both collection periods using SINE PCR, revealed that all of them were *An. gambiae* s.s (Appendix A). Overall, 8/195 (4.10%) mosquitoes were infected with *Plasmodium* for the two collection periods. In November, 2/94 (2.12%) mosquitoes were found infected with *Plasmodium falciparum* but none with *Plasmodium ovale*, *vivax* or *malariae*. In May, 4/101 (3.96%) and 2/101 (1.98%) mosquitoes were, respectively, infected by *Plasmodium falciparum* and *malariae*. No significant difference was found between the infection rates according to the season (*p*-value > 0.05).

### 3.2. Agrochemical’s Residues Concentration in Mosquitoes Breeding Water and Soil

In the soil sample collected in IRAD and Djincha, we found no traces of deltamethrin residues (Appendix A). In the mosquito breeding water, we noticed the absence of deltamethrin residues in the two locations. However, we observed a higher amount of alpha-cypermethrin in water collected in IRAD with a concentration higher than the LOD of the method (2.903 µg/mL = 5.16 fold LOD) compared to Djincha with concentration of 0.443 µg/mL (0.78 fold LOD) (Table 1).

### 3.3. Insecticide Resistance Profile

#### 3.3.1. Susceptibility Profile and Resistance Intensity

F_1_ progeny from Mangoum field-collected female *An. gambiae* s.s. mosquitoes were resistant to permethrin (6.58 ± 0.25%), alpha-cypermethrin (1.18 ± 0.08%), deltamethrin (0.02 ± 0.21%), bendiocarb (27.50 ± 0.35%), propoxur (33.33 ± 0.55%), DDT (1.18 ± 0.08%) and malathion (53.62 ± 0.44%) (Figure 1). Due to the high level of resistance observed to pyrethroids, intensity bioassays were carried out with 5× diagnostic concentration (DC) and 10× DC of permethrin (3.75% and 7.5%), alpha-cypermethrin (0.25% and 0.5%) and deltamethrin (0.25% and 0.5%). A high intensity of resistance to permethrin (5×: 58.51 ± 0.61%; 10×: 93.27 ± 0.21%), alpha-cypermethrin (5×: 63.88 ± 0.44%; 10×: 83.10 ± 0.31%) and deltamethrin (5×: 29.87 ± 0.46%; 10×: 37.23 ± 0.31%) were found (Figure 2). We also assessed seasonal variation in permethrin susceptibility. A significant effect of collection period on the susceptibility of *An. gambiae* s.s. to permethrin was observed in F_0_ mosquitoes coming from larval collection (*p*-value < 0.05), but not with the F_1_ coming from indoor aspiration (Figure 2A). No significant difference was found between F_0_ and F_1_ in November collection, but in May, there was a significant difference between F_0_ and F_1_ with permethrin 1× in May collection (*p*-value < 0.01).

#### 3.3.2. Synergist Bioassay with PBO

To assess the implication of the cytochrome P450s in the resistance observed to permethrin, alpha-cypermethrin and deltamethrin, mosquitoes collected from Mangoum were pre-exposed to PBO then to permethrin, alpha-cypermethrin or deltamethrin. A partial recovery of the susceptibility was observed after pre-exposure to the PBO (mortality: 15.48 ± 1.20%, 23.38 ± 0.84% and 45.65 ± 1.49% respectively) compared to the result of 1× permethrin, 1× alphacy-permethrin and 1× deltamethrin alone (mortality: 6.58 ± 0.25%, 1.18 ± 0.08% and 0.02 ± 0.21%, respectively) (Figure 1).

### 3.4. Bioefficacy of Insecticide-Treated Bed Nets

Very-low efficacy of standard pyrethroid-only nets was also observed against *An. gambiae* s.s.: no mortality for Olyset and DuraNet, respectively, and 4.0 ± 0.17% for PermaNet 2.0. However, PBO-based nets (Olyset Plus, and PermaNet 3.0) showed an increased efficacy (Olyset Plus: 92.0 ± 0.26% mortality; PermaNet 3.0-roof: 100.0 ± 0.0%) (Figure 3). The mortality with PermaNet 3.0 side did not differ from that of PermaNet 2.0 (4.0 ± 0.17% vs. 4.08 ± 0.28%) in Mangoum, indicating the high intensity of resistance in this location. Pyrethroid-only and PBO-nets used in this study induced total mortality against the control Kisumu susceptible *An. gambiae* s.s mosquitoes. Mangoum mosquitoes were also sensitive to nets impregnated with pyrethroids coupled with pyriproxyfen (Royal guard: 84.78 ± 0.68% mortality). We found that the mortality rate of Interceptor was higher than Interceptor G2 (18.75 ± 0.56% vs 5.0 ± 0.17%) (Figure 3).

### 3.5. Target-Site Resistance Markers in An. gambiae from Mangoum

In total, 48 oviposited F_0_ females *An. gambiae* s.s from Mangoum were genotyped for target-site resistance markers. The *1014F-kdr* resistant allele was found to be fixed in Mangoum with 100% (48/48) homozygotes RR. All samples were homozygote SS for the *L1014S* genotyping. Out of 44 oviposited female mosquitoes which successfully amplified for the *N1575Y-kdr* mutation, 11 were heterozygote resistant (25%) and 33 were homozygote susceptible (75%). The allelic frequency of the resistant *G119S* mutation was 51.1% (23/45 RR and 22/45 SS) (Figure 4). No significant association was found between the *1575Y-kdr* and the ability to survive exposure to permethrin 5× and Deltamethrin 10×. However, mosquitoes having the 1575Y-kdr allele had four times more chance to survive deltamethrin 5× exposure (OR: 4.01; 95% Confidence interval: 1.29–13.90; *p*-value: 0.03) (Table 2). To assess the genetic diversity and detect potential signatures of selection acting on the voltage-gated sodium channel, a 498-bp portion of this gene spanning exon 20 and the 1014 codon was sequenced in 15 *An. gambiae* s.s. from Mangoum. Analysis revealed a signature of selection in the VGSC in *An. gambiae s.s.* from Mangoum with a lack of genetic diversity shown by a single predominant haplotype (Figure 5). Comparison of the Mangoum-VGSC haplotype with six *kdr*-bearing haplotypes previously detected across Africa revealed that the 1014F haplotype found in Mangoum belong to the H3-1014F haplotype, predominant in West/Central Africa [30].

### 3.6. Transcriptional Profiling of Metabolic Resistance Genes in An. gambiae s.s.

The expression level of *CYP4G16*, *CYP4G17*, *CYP6M2*, *CYP6Z1*, *CYP6Z2*, *GSTe2*, *CYP9K1*, *CYP6P1*, *CYP6P3*, *CYP6P4*, *SAP1*, *SAP2* and *SAP3* was evaluated in *An. gambiae* from Mangoum relative to the laboratory strain (Kisumu) and using two housekeeping genes (*EF* and *RSP7*). For the permethrin-exposed mosquito populations, we noted the overexpression of four metabolic genes (*CYP6M2* (10-fold), *CYP6Z1* (17-fold), *CYP6Z2* (13-fold), and *CYP9K1* (13-fold)), two cuticular resistance encoding genes (*CYP4G16* (6-fold) and *CYP4G17* (5-fold)) and tree sensory appendage protein encoding genes (*SAP1* (28-fold), *SAP2* (5-fold) and *SAP3* (24-fold)) (Figure 6A). However, we did not observe any significant difference in the expression of these genes between the unexposed and exposed (*p*-value > 0.05). For deltamethrin-exposed mosquitoes, overexpression was also found in four metabolic genes (*CYP6M2* (34-fold), *CYP6Z1* (29-fold), *CYP6Z2* (65-fold), and *CYP9K1* (87-fold)), two cuticular resistance encoding genes (*CYP4G16* (8-fold) and *CYP4G17* (27-fold)) and tree sensory appendage protein encoding genes (*SAP1* (13-fold), *SAP2* (5-fold) and *SAP3* (8-fold)) (Figure 6B). There was a significant difference between the expression levels of *CYP6P3* and *CYP4G17* between the exposed and unexposed groups respectively. However, this difference was not observed between populations of mosquitoes exposed to increasing doses of insecticide (1×, 5× and 10×).

## 4. Discussion

There is growing evidence of the emergence of high resistance to pyrethroids in major malaria vectors including *An. gambiae* leading to the low efficiency of commonly used pyrethroid-only bed-net [8,13]. It is thus urgent to establish the magnitude of resistance aggravation and to investigate the potential drivers of such escalation in natural populations of malaria vectors. Some studies demonstrated that mosquito larvae recurrently exposed to agricultural pesticides could develop resistance against public health insecticide classes and could constitute a factor of resistance aggravation [38]. Herein, we aimed to assess resistance intensity, and highlight the molecular drivers of the resistance escalation in the main malaria vector *An. gambiae* and their impact on control tools in an agricultural hotspot such as Mangoum in Cameroon.

### 4.1. An. gambiae Is Driving Malaria Transmission in a Context of Intense Agricultural Activities

*An. gambiae* s.s. was the only species found in Mangoum. Surrounded by the river Noun, the locality of Mangoum is intensely irrigated and constitutes a site of predilection for market gardening. Tomato cultivation is the predominant agricultural activity, along with corn and beans. The creation of numerous watering holes used to water the fields is the major cause of the stagnation of the Noun’s water flowing in this locality. The constant presence of retention water in the fields and swamp makes this locality a particularly risky area for the proliferation of *An. gambiae* species, as observed in our study. Moreover, this risk is all the more alarming since some of these sites are subject to a high selective pressure due to the massive use of certain pesticides containing, among others, alpha-cypermethrin, also used as active ingredient in many LLINs. Despite the presence of *An. gambiae* adults in the households, we noted a low frequency of *Plasmodium* infection in these mosquitoes. The low *Plasmodium* infection rate found in *An. gambiae* from Mangoum was lower than the infection rate found by Atangana et al. (2010) [39], ten years before the implementation of LLINs as a vector control strategy in Mangoum. They found that infection rate was 8.9% in 2010 compared to 4.10% found in the present study. Many reasons could explain this difference. First of all, the role played by the LLINs intervention. It is known that LLINs play a double role by protecting humans from mosquito bites and by killing mosquitoes that encounter the net. In addition, the method used by Antangana and colleagues was ELISA CSP looking for the *Plasmodium* protein in the salivary gland of mosquitoes. This technique could show a high level of false positives, particularly in the case of zoophilic blood feeding [40]. The present study did not establish a correlation between sporozoite infectivity and insecticide resistance. Further studies to demonstrate the relationship between sporozoite infectivity and insecticide resistance in Mangoum are needed.

### 4.2. An. gambiae in Mangoum Exhibits High Resistance to the Four Classes of Insecticides

Very low mortality rate was observed in *An. gambiae* from Mangoum mosquito population against the discriminating concentrations of pyrethroids but also the other three classes of insecticides (Carbamate, Organophosphate and Organochlorine). This is among the first studies in Cameroon reporting resistance to all the four classes of insecticides commonly used in public health. Many studies reported escalation in resistance to pyrethroids across the country in *An. coluzzii* [29,41] and *An. funestus* [42] but it is the first time resistance have been noticed for organophosphates. The high intensity of pyrethroid resistance observed in *An. gambiae* from Mangoum was also reported in *An. gambiae* s.l. and *An. funestus* across the continent [8,10,13,42]. This multiple and high level of resistance has drastic consequences on malaria control tools. As previously mentioned in other countries [13,43], the efficacy of insecticidal treated LLINs in *An. gambiae* s.s from Mangoum were significantly reduced. However, PBO-based net (Olyset plus^®^ and PermaNet^®^ 3.0) induced higher mortality in this population showing that these second-generation LLINs could be suitable for vector control in this locality. This greater efficacy of PBO-based LLINs further supports the contribution of cytochrome P450s enzymes in the high resistance level observed in Mangoum. However, WHO tube tests showed a partial recovery of susceptibility when exposed to PBO, suggesting that, beside P450s, other mechanisms are potentially involved [44]. The same result has been obtained in Kedougou (Senegal) [8]. Furthermore, our results showed a loss efficacy of the new generation bed net Interceptor G2^®^ in Mangoum with very low mortality 72 h post-exposure to this net. This reduced efficacy of new generation bed nets could be linked, first, to the widespread use of alpha-cypermethrin based net and pesticides in this locality [45]. Nevertheless, in this study, we found residual alpha-cypermethrin in the water coming from mosquito’s breeding sites in Mangoum. It could be a result of the intensive use of insecticides from agricultural activities. It is worth mentioning that during soil and water collection in farms, we also noticed the presence of many pyrethroid-based packaging materials such as containers suggestive of the intensive use of such pesticides in the collection sites, especially in IRAD. This could justify the higher concentration of insecticide residues found in breeding water from that site compared to Djincha, where agricultural activity is moderate. The presence of insecticide residues precisely in breeding water where the larval stages of mosquitoes develop, must have built an increased and permanent insecticide pressure driving the escalation of resistance observed in this study. The absence of insecticide residues in soil as revealed by our study is in line with a previous report by Pal and colleagues [46]. Insecticides have been shown to degrade faster in soil due to the physicochemical properties and microbial biomass of the soil [46]. Although in this study the presence in water and soil of type I pyrethroid-based pesticides was not tested, that does not mean they are not being used in the study site. They could, as well, be contributing to insecticide pressure involved in the observed resistance escalation. Another justification for the reduced efficacy of new generation bed nets could be the inadequacy of cone assays to assess efficacy of slow acting insecticides such as chlorfenapyr. Normally, Chlorfenapyr has a reputation for slow action and ‘delayed’ toxic activity 2–3 days post-exposure. It will be good to try experimental huts and or tunnel assays to check blood feeding inhibition and repellency.

### 4.3. Molecular Drivers of Resistance Escalation Are Likely Complex Combining Several Mechanisms

In Mangoum, the *1014F* resistant allele was fixed as previously reported on the continent [13,47]. Polymorphic analysis of the exon 20 of the VGSC gene showed that the fixation of 1014F allele is associated with reduced genetic diversity with a predominant haplotype as reported in several studies [29,33]. This is evidence of extensive selection on this population in Mangoum, likely from both LLINs use and agricultural practices. Our results showed a significant association between the *N1575Y* mutation and pyrethroid resistance. Available data showed that the *1575Y* allele conferred to mosquitoes the ability to survive to 5× deltamethrin exposure, suggesting that the increased frequency of this allele could be contributing to resistance escalation. It had been found elsewhere that *N1575Y* could compensate for the fitness cost incurred by the *L1014F kdr* mutation and provide additional pyrethroid resistance [48]; therefore, the frequency of this allele should be further monitored as an indicator of resistance aggravation.

The results of the expression of candidate genes involved in metabolic, cuticular and sensory resistance revealed the over-expression of several cytochrome P450s, notably, *CYP9K1*, *CYP6M2*, *CYP6Z1* and *CYP6Z2*, which is similar to previous studies highlighting the contribution of P450 genes to resistance development in *An. gambiae* [49], *An. coluzzii* [29] and *An. funestus* [13]. Similarly, the over-expression of other gene families point to the fact that the multiple resistance observed in Mangoum is multifactorial, supporting the partial recovery from PBO synergist assays. Similar observations have been made in multiple resistant populations, such as the ones reported in Tiassale [49] in Ivory Coast or in Vallee du Kou in Burkina Faso [5,50]. However, no significant difference in expression was observed between mosquitoes surviving different insecticide doses, suggesting that the underlying molecular mechanisms of this resistance escalation remain unclear. This is similar to work performed in *An. funestus* in Uganda, where the major resistance P450 gene *CYP9K1* was highly expressed but equally in all samples [13]. The lack of correlation between the expression of major genes and resistance intensity suggests that the underlying mechanisms remain to be elucidated. However, the fact that some resistance alleles are already fixed, such as kdr, could make it difficult to assess their contribution to resistance escalation using these field samples. It could be useful to consider other strategies, including the use of crossing, to decipher what molecular factor contributes the most to the growing resistance escalation in natural populations of malaria vectors.

## 5. Conclusions

The extremely high intensity of resistance coupled with the loss in efficacy of impregnated bed nets against *An. gambiae* s.s. in Mangoum represents a serious threat for vector control. PBO-based nets could be used as an alternative measure to sustain malaria control in the study area. Multiple metabolic resistance mechanisms were found to be involved in resistance. However, the common P450-based mechanisms were not found to play a role in this resistance escalation highlighting the need for alternative approaches to elucidate the main molecular drivers of resistance aggravation.

## Figures and Tables

**Figure 1 genes-13-01206-f001:**
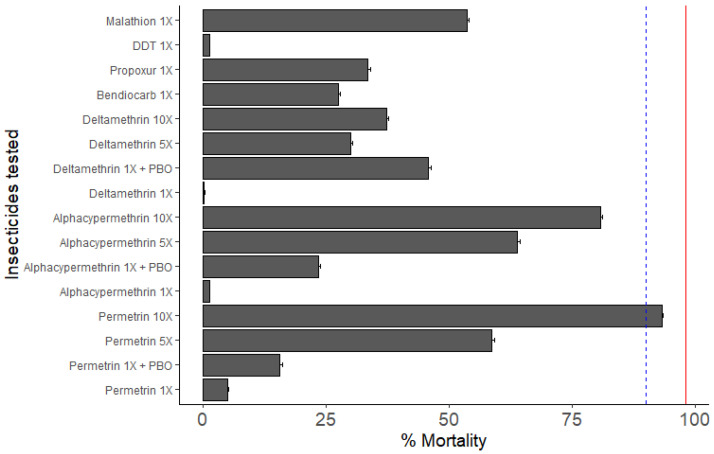
Susceptibility profile of *An. gambiae* s.s. to insecticides and effect of pre-exposure to synergist PBO against permethrin, alpha-cypermethrin and deltamethrin. Recorded mortalities following 60-minute exposure of *An. gambiae* s.s. from Mangoum to different insecticides. Data are shown as mean ± standard error of the mean (SEM). Blue line corresponds to 80% mortality and red line to 98% mortality.

**Figure 2 genes-13-01206-f002:**
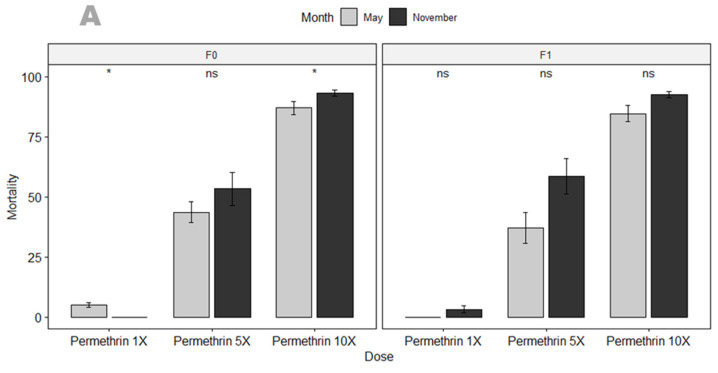
(**A**) Temporal variation in susceptibility profile of *An. gambiae* s.s. (F_0_ & F_1_) to different doses of permethrin. (**B**) Susceptibility profile of F_0_ and F_1_
*An. gambiae* s.s. to different doses of permethrin per collection time point (May & November). Recorded mortalities following 60-minute exposure of *An. gambiae* s.s. from Mangoum. Data are shown as mean ± standard error of the mean (SEM). Student’s test was used for comparisons. *p* value code: ns: *p* > 0.05. *: *p* ≤ 0.05, **: *p* ≤ 0.01.

**Figure 3 genes-13-01206-f003:**
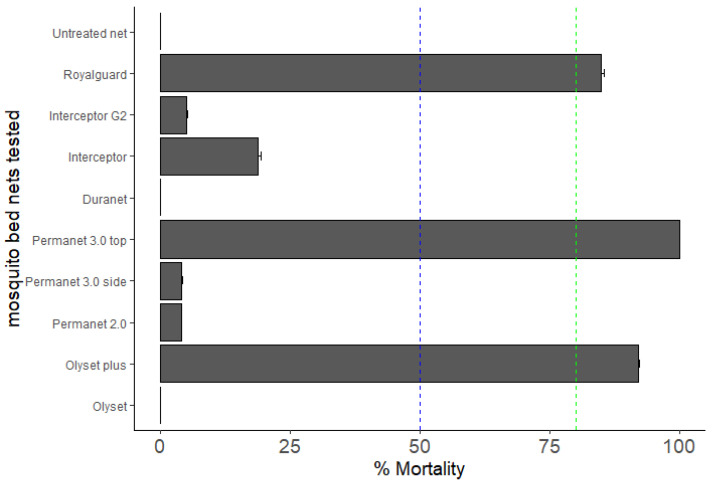
Bio-efficacy of different commercial LLINs against *An. gambiae* s.s. in Mangoum. Results of cone bioassays with Olyset^®^Net, Olyset^®^Plus, PermaNet^®^2.0, PermaNet^®^3.0 (side and roof), Duranet^®^, Royal guard^®^, Inteceptor^®^ and Interceptor G2^®^ (% Mortality 72 h). Results are average of percentage mortalities ± SEM of five replicates. Mortality < 50% (blue line): No efficient, 50% < Mortality ≤ 80%: minimal efficacy, Mortality ≥ 80% (green line): optimal efficacy.

**Figure 4 genes-13-01206-f004:**
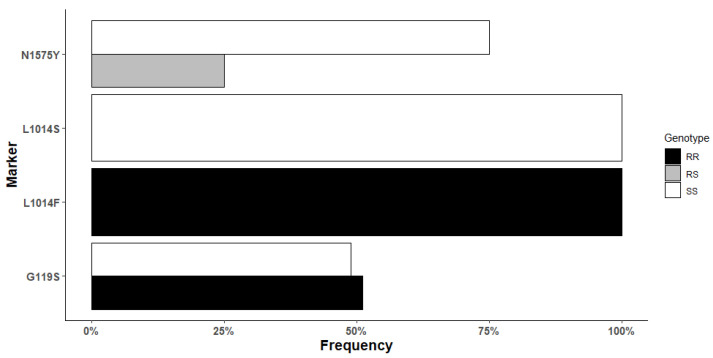
Genotyping of resistance markers in F_0_
*An. gambiae* s.s. from Mangoum.

**Figure 5 genes-13-01206-f005:**
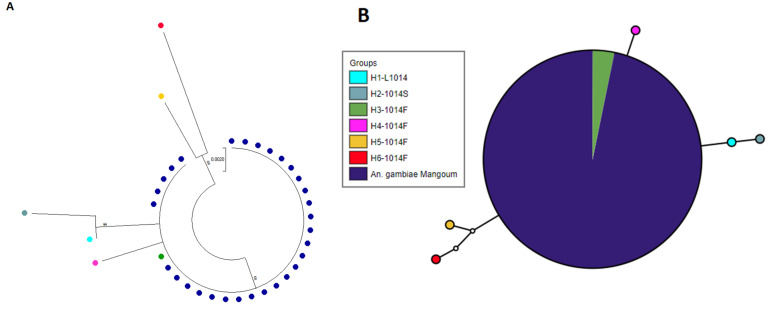
Analysis of the polymorphism of a portion of the voltage-gated sodium channel (VGSC) gene spanning the L1014F/S mutation. (**A**) Maximum likelihood phylogenetic tree of VGSC fragment with previously recorded 1014F/S haplotypes across Africa. (**B**) Templeton–Crandall–Singh network for the VGSC haplotypes in F_0_ mosquitoes from Mangoum.

**Figure 6 genes-13-01206-f006:**
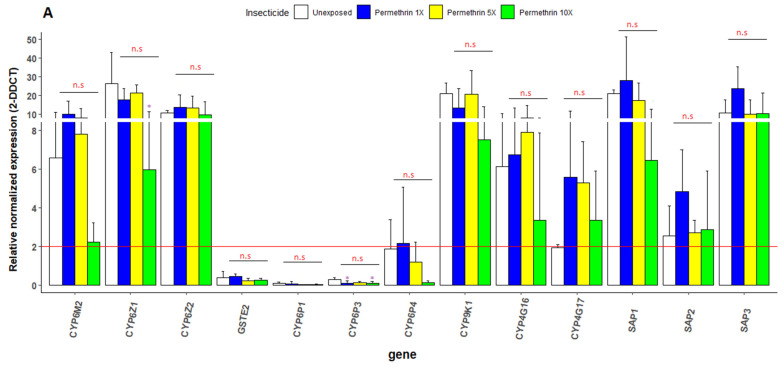
Differential expression by quantitative reverse-transcription polymerase chain reaction of the major insecticide resistance genes in *An. gambiae* in Mangoum compared with the susceptible Kisumu. (**A**) Permethrin, (**B**) deltamethrin. Error bars represent standard error of the mean at 95% confidence interval, with significance * *p* ≤ 0.05, ** *p* ≤ 0.01 as calculated by Kruskal–Wallis test for between exposed groups comparisons (in red) and Dunnett’s test for comparing several each exposed groups with a unexposed (in purple). The red line represent the two-fold change threshold. n.s: non significant.

**Table 1 genes-13-01206-t001:** Agrochemical’s residues concentration and linearity parameters.

	Alpha-Cypermethrin LOD = 0.563; LOQ = 52.918		Deltamethrin LOD = 1.899; LOQ = 5.754	
	Breeding Water	Soil Sediment	Breeding Water	Soil SEDIMENT
	µg/mL	µg/mL	µg/mL	µg/mL
IRAD	2.903 ± 0.06	ND	ND	ND
Djincha	0.443 ± 0.002	ND	ND	ND

LOD, limit of detection; LOQ, limit of quantification; ND, not detected.

**Table 2 genes-13-01206-t002:** Allelic frequency of 1575Y-kdr between alive and dead mosquitoes from Mangoum exposed to pyrethroids.

Insecticides	Alive			Death			OR (95% CI)	*p* Value
	NY	NN	Frequency (%)	NY	NN	Frequency (%)		
Permethrin 5×	10	20	16.67%	12	18	20.00%	0.75 (0.25–2.19)	0.79
Deltamethrin 5×	14	14	25.00%	6	25	9.68%	4.01 (1.29–13.90)	0.03
Deltamethrin 10×	11	17	19.64%	10	21	16.13%	1.35 (0.46–4.04)	0.6

## Data Availability

The datasets generated and/or analyzed during the current study are available from the corresponding author upon request.

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
