# Peer review of "Molecular Drivers of Multiple and Elevated Resistance to Insecticides in a Population of the Malaria Vector Anopheles gambiae in Agriculture Hotspot of West Cameroon"

_genes, 2022, doi:10.3390/genes13071206_

Round 1
Reviewer 1 Report
The manuscript titled “Molecular driv4rs of multiple and elevated resistance to insecticides in a population of the malaria vector Anopheles gambiae in agriculture hotspot of we Cameroon” (Genes-1724989) is being considered as a research article. Overall, the manuscript is a set of nice studies investigating the molecular mechanisms of mosquitoes obtained from the wild in area with agriculture production. The approach is sound, and the conclusions are supported by the results. There are several, grammar, punctuation, and spelling errors that should be correct, some of which are stated below.
Line number |
Comment |
22 |
Rephase sentence does not make sense |
26 |
Emerged instead of emerging |
56 |
Rephrase |
58 |
Reduction should not be plural |
59 |
There is more than one pyrethroid, therefore, it should be plural then replace “is” with “are” |
61-62 |
Why is this an isolated sentence and not part of a paragraph |
70 |
10 fold should be hyphenated and extra space after “in” and “noted” |
97-98 |
This first sentence is arguable incorrect. These studies have been performed? The authors could be more specific/ |
155 |
Please provide the purity of the deltamethrin and alpha-cypermethring along with the company they were obtained from |
205 |
Abbott’s formula is common unless there was high mortality in the control. At what level of mortality in the control were the parallel experiments not used (i.e., 20%). Please provide this information |
294-295 |
Not clear why a single sentence is alone and not included in a paragraph. |
Figure 1 |
Arguably not necessary or used as a supplemental figure |
349 |
Hyphen needed in alphacypermethrin – here and throughout the manuscript |
375 |
Target site should be hyphenated, here and throughout |
477 |
Hyphen needed in post-exposure |
Author Response
Line number |
Comment |
Response to Reviewer 1 comment |
22 |
Rephase sentence does not make sense |
Done |
26 |
Emerged instead of emerging |
Changed |
56 |
Rephrase |
Done |
58 |
Reduction should not be plural |
Corrected |
59 |
There is more than one pyrethroid, therefore, it should be plural then replace “is” with “are” |
Corrected |
61-62 |
Why is this an isolated sentence and not part of a paragraph |
It was a mistake. We linked it to the previous paragraph |
70 |
10 fold should be hyphenated and extra space after “in” and “noted” |
Corrected |
97-98 |
This first sentence is arguable incorrect. These studies have been performed? The authors could be more specific/ |
Corrected |
155 |
Please provide the purity of the deltamethrin and alpha-cypermethring along with the company they were obtained from |
Done |
205 |
Abbott’s formula is common unless there was high mortality in the control. At what level of mortality in the control were the parallel experiments not used (i.e., 20%). Please provide this information |
The level of control mortality required for the used of Abott's formula was provide in the manuscript. |
294-295 |
Not clear why a single sentence is alone and not included in a paragraph. |
We linked this sentence to the following paragraph |
Figure 1 |
Arguably not necessary or used as a supplemental figure |
This figure was sent to supplemental material. |
349 |
Hyphen needed in alphacypermethrin – here and throughout the manuscript |
Correction done |
375 |
Target site should be hyphenated, here and throughout |
Correction done |
477 |
Hyphen needed in post-exposure |
Correction done |
|
|
|

Reviewer 2 Report
The submitted manuscript Genes-1724989 has done lot of experimental work but did not present the data in a concise and standard way. I found that information is discrete and repetitive. It is very difficult to any one to point out every mistake, here are few examples-
In methods- there are 12 method sections; 2.1 with 2.2 and 2.3 with 2.4 can be merged. So many things are repetitive like use of WHO insecticidal bed etc.
There are two figures named as 1 in the manuscript, and both are not required, you can place them in supplementary information. Figure 2 should be figure 1 in my opinion, as this is the main storyteller. Figure 7 is missing.
Why only the sporozoite infectivity levels were tested, other Plasmodium stages can be a found in field collected mosquitoes, oocyst or ookinete. There is no relationship of information like sporozoites infectivity rate and insecticide resistance and why it is important to show the data with month and year.
Provide all the primer information used for multiple purposes (species identification, sporozoites infectivity and insecticide resistance gene etc.) in the supplementary file.
For field collected mosquitoes species identification at molecular level, you must present the data in supplementary file, as given in PMID: 27353585 (J Vector Borne Dis. 2016;53(2):149-55).
Table 1 column 1 has two different parameters; why LOD and LOQ are placed in same place.
Quality of figure should be improved. In figure number 8 (currently), make the break and increase the y-axis level, to show all the data points and gene expression. Axis title and font should be prominently visible.
A good quality of scientific figure is always enlightened by the statistically significant values. Except table 2, I did not find any other places a good statistical presentation. As you have shown multiple experiments with variable concentration of insecticides and relative gene expression hence it is must to show the p values.
There are plenty of these mistakes hence I would suggest describing the methods precisely, but cover everything one after another. Brush up the figures and table and revise the manuscript one more time including English language.
All the very best.
Author Response
Point 1: In methods- there are 12 method sections; 2.1 with 2.2 and 2.3 with 2.4 can be merged. So many things are repetitive like use of WHO insecticidal bed etc.
Response 1: 2.1 with 2.2 and 2.3 with 2.4 have been merged in the revised manuscript and the number of method sections adjusted accordingly.
Point 2: There are two figures named as 1 in the manuscript, and both are not required, you can place them in supplementary information. Figure 2 should be figure 1 in my opinion, as this is the main storyteller. Figure 7 is missing.
Response 2: This suggestion has been taken into account accepted. Figure 1 was send to supplementary information. Figure labels were corrected in the manuscript.
Point 3: Why only the sporozoite infectivity levels were tested, other Plasmodium stages can be a found in field collected mosquitoes, oocyst or ookinete. There is no relationship of information like sporozoites infectivity rate and insecticide resistance and why it is important to show the data with month and year.
Response 3: We chose to look only for the presence of sporozoites because they are infective stage parasites as correlate with the likehood of the mosquito transmitting the parasite to the human host. The presence of oocysts or ookinetes does not necessarily guarantee that they will reach the sporozoite stage and be transmitted to the host. The purpose of this analysis was to characterize our study population in order to give an alert on the disease transmission potential of the mosquitoes present in our study site. However, no correlation test was done between sporozoite infectivity and insecticide resistance as this was not the focus of this study.
Considering that the samples used for the evaluation of gene expression were selected over different periods and seasons of collection, it was important to present the resistance profile for both collection periods taking into account the seasonial variation in resistance in mosquitoes.
Point 4: Provide all the primer information used for multiple purposes (species identification, sporozoites infectivity and insecticide resistance gene etc.) in the supplementary file.
Response 4: All the primer information used in this study has been provided as a supplementary file
Point 5: For field collected mosquitoes species identification at molecular level, you must present the data in supplementary file, as given in PMID: 27353585 (J Vector Borne Dis. 2016;53(2):149-55).
Response 5: Figure S2 was provided in supplementary file
Point 6: Table 1 column 1 has two different parameters; why LOD and LOQ are placed in same place.
Response 6: Thanks, we remove LOD and LOQ from column 1 and reset the table 1 as found in the corrected version.
Point 7: Quality of figure should be improved. In figure number 8 (currently), make the break and increase the y-axis level, to show all the data points and gene expression. Axis title and font should be prominently visible.
Response 7: Correction done.
Point 8: A good quality of scientific figure is always enlightened by the statistically significant values. Except table 2, I did not find any other places a good statistical presentation. As you have shown multiple experiments with variable concentration of insecticides and relative gene expression hence it is must to show the p values.
Response 8: We increase the quality of figure 6. The p values had been included and the title was rephrase. For the figure 1 and 3, we did not included p value, but we showed a threshold according to the WHO guidelines.

Round 2
Reviewer 2 Report
I appreciate that author has done the substantial changes in the the present manuscript and answered the reviewer's questions properly. In the track change version it is difficult for me to assess it completely.
I would recommend that in the supplementary file in primer list table, please add the column for amplicon size purpose for better understanding and one can follow that easily. You can take the help of this and previous article and can cite also https://doi.org/10.1016/j.gene.2016.09.022
All the best.
Author Response
Comment 1: I would recommend that in the supplementary file in primer list table, please add the column for amplicon size purpose for better understanding and one can follow that easily. You can take the help of this and previous article and can cite also https://doi.org/10.1016/j.gene.2016.09.022
Response 1: Thank you for the recommendation, We corrected the supplementary file
Comment 2: Moderate English changes required
Response 2: We also made some English corrections that could be followed in the tracking version.